# Serum miR-192-5p levels predict the efficacy of pegylated interferon therapy for chronic hepatitis B

Yoshihito Nagura[1,2], Kentaro Matsuura[2], Etsuko Iio[1], Koji Fujita[3], Takako Inoue[4], Akihiro Matsumoto[5], Eiji Tanaka[5], Shuhei Nishiguchi[6], Jong-Hon Kang[7], Takeshi Matsui[7], Masaru Enomoto[8], Hiroki Ikeda[9], Tsunamasa Watanabe[9], Chiaki Okuse[10], Masataka Tsuge[11], Masanori Atsukawa[12], Masakuni Tateyama[13], Hiromi Kataoka[2], Yasuhito Tanaka[1,13]*

1 Departments of Virology & Liver Unit, Nagoya City University Graduate School of Medical Sciences, Nagoya, Aichi, Japan, 2 Department of Gastroenterology and Metabolism, Nagoya City University Graduate School of Medical Sciences, Nagoya, Aichi, Japan, 3 Department of Gastroenterology and Neurology, Faculty of Medicine, Kagawa University, Miki, Kagawa, Japan, 4 Department of Clinical Laboratory Medicine, Nagoya City University Hospital, Nagoya, Aichi, Japan, 5 Department of Internal Medicine, Shinshu University School of Medicine, Matsumoto, Nagano, Japan, 6 Division of Hepatobiliary and Pancreatic Disease, Department of Internal Medicine, Hyogo College of Medicine, Nishinomiya, Hyogo, Japan, 7 Division of Center for Gastroenterology, Teine Keijinkai Hospital, Sapporo, Hokkaido, Japan, 8 Department of Hepatology, Graduate School of Medicine, Osaka City University, Osaka, Osaka, Japan, 9 Division of Gastroenterology and Hepatology, Department of Internal Medicine, St. Marianna University School of Medicine, Kawasaki, Kanagawa, Japan, 10 Division of General Internal Medicine, Department of Internal Medicine, Kawasaki Municipal Tama Hospital, Kawasaki, Kanagawa, Japan, 11 Department of Gastroenterology and Metabolism, Graduate School of Biomedical and Health Sciences, Hiroshima University, Hiroshima, Hiroshima, Japan, 12 Division of Gastroenterology and Hepatology, Department of Internal Medicine, Nippon Medical School, Tokyo, Japan, 13 Department of Gastroenterology and Hepatology Faculty of Life Sciences, Kumamoto University, Kumamoto, Kumamoto, Japan

* ytanaka@kumamoto-u.ac.jp

**Data Availability Statement:** All relevant data are within the paper and its Supporting information files.

## Abstract

We examined the association between serum miRNA (-192-5p, -122-3p, -320a and -6126-5p) levels and the efficacy of pegylated interferon (Peg-IFN) monotherapy for chronic hepatitis B (CHB) patients. We enrolled 61 CHB patients treated with Peg-IFNα-2a weekly for 48 weeks, of whom 12 had a virological response (VR) and 49 did not VR (non-VR). A VR was defined as HBV DNA < 2,000 IU/ml, hepatitis B e antigen (HBeAg)-negative, and nucleos(t)ide analogue free at 48 weeks after the end of treatment. The non-VR group showed a significantly higher HBeAg-positivity rate, ALT, HBV DNA, and serum miR-192-5p levels at baseline ($P = 0.024$, $P = 0.020$, $P = 0.007$, $P = 0.021$, respectively). Serum miR-192-5p levels at 24-weeks after the start of treatment were also significantly higher in the non-VR than the VR group ($P = 0.011$). Multivariate logistic regression analysis for predicting VR showed that miR-192-5p level at baseline was an independent factor (Odds 4.5, $P = 0.041$). Serum miR-192-5p levels were significantly correlated with the levels of HBV DNA, hepatitis B core-related antigen, and hepatitis B surface antigen (r = 0.484, 0.384 and 0.759, respectively). The serum miR-192-5p level was useful as a biomarker for the therapeutic efficacy of Peg-IFN in CHB treatment.

**Funding:** Y.T was supported by a grant-in-aid from the Research Program on Hepatitis from the Japan Agency for Medical Research and Development (AMED JP20fk0310101, JP21fk0310101). URL: https://www.amed.go.jp/ The funders had no role in study design, data collection and analysis, decision to publish, or preparation of the manuscript.

**Competing interests:** Regarding COI of Yasuhito Tanaka: Research funding from C Fujifilim Corp., Janssen Pharmaceutical K.K, Gilead Sciences, GlaxoSmithKline Pharmaceuticals Ltd, and Stanford Junior University, and lecture fees from Fujirebio, Inc. and Gilead Sciences, these do not relate to employment, consultancy, patents, products in development, and marketed products. Therefore, these do not alter our adherence to PLOS ONE policies on sharing data and materials.

## Introduction

Hepatitis B virus (HBV) infection is a global public health problem, with approximately 240 million people, or 6% of the world's population, chronically infected with HBV [1]. The prevalence of hepatitis B is highest in the Western Pacific and African regions, affecting 5% to 7% and > 8% of the adult population, respectively [2].

The long-term goal of antiviral therapy for chronic hepatitis B (CHB) has been to eliminate hepatitis B surface antigen (HBsAg). However, the current standard therapies using nucleos(t) ide analogues (NAs) or pegylated interferon (Peg-IFN) are difficult to achieve the elimination of HBsAg [3]. Nucleos(t)ide analogues have been shown to be highly safe across a wide range of patients with CHB, including decompensated cirrhosis and pregnancy [4]. However, recurrence of elevated levels of HBV DNA and alanine aminotransferase (ALT) is likely when NAs treatment is discontinued, therefore long-term administration of NAs is often required [5]. Pegylated interferon is thought to be more effective than lamivudine based on HBV DNA suppression and seroconversion of HBsAg antibody [6]. Furthermore, Peg-IFN has the advantage of maintaining a drug-free therapeutic effect without additional drug administration after treatment in CHB patients who exhibit a therapeutic response [7, 8]. However, the therapeutic effect of Peg-IFN is obtained in only 20% to 30% of patients who are hepatitis B e antigen (HBeAg)-positive and in 20% to 40% of HBeAg-negative patients [6]. Additionally, Peg-IFN therapy is associated with various side effects such as fever, fatigue, depression, neutropenia and thrombocytopenia [7]. The levels of ALT and HBV DNA, and HBV genotype at baseline, have been reported to significantly affect the response to Peg-IFN therapy after 24-weeks of treatment [8–10]. Along these lines, a reliable marker for the efficacy of Peg-IFN therapy in CHB is needed [11].

MicroRNAs (miRNAs) are involved in various biological phenomena, such as cell development, differentiation, proliferation, apoptosis, and metabolism and also play roles in the pathogenesis of inflammation, fibrogenesis, and carcinogenesis in liver diseases [12, 13]. Several studies to date have revealed an association between serum miRNA levels and the response to IFN therapy in CHB. Brunetto et al. showed that the levels of several serum miRNAs such as miR-192-5p, miR-320a, and miR-122-3p were related to the response to IFN therapy in CHB [14]. A study by Fujita et al. revealed that higher miR-6126-5p levels in sera during Peg-IFN therapy with or without NAs for CHB predicted the reduction of HBsAg after the completion of therapy [15]. However, these studies analyzed only a relatively small numbers of patients. Therefore, the association between serum expression levels of these miRNAs and the response to IFN therapy in CHB should be validated in independent cohorts.

Herein, we aimed to validate the association between the levels of serum miR-192-5p, 320a, 122-3p, and 6126-5p with the efficacy of Peg-IFN monotherapy for CHB patients.

## Materials and methods

### Patients and study design

The design of this retrospective study is shown in Fig 1. We enrolled 61 CHB patients from 2012 to 2016 in 8 hospitals (Nagoya City University Hospital, Shinshu University Hospital, Hyogo College of Medicine Hospital, Osaka City University Hospital, Chiba University Hospital, St. Marianna Medical University Hospital, Hiroshima University Hospital, and Nippon Medical School Chiba Hokusoh Hospital). All patients were chronically infected with HBV and confirmed to be HBsAg-positive for at least 6 months. Patients with a history of hepatocellular carcinoma, cirrhosis, other causes of liver disease such as autoimmune hepatitis and

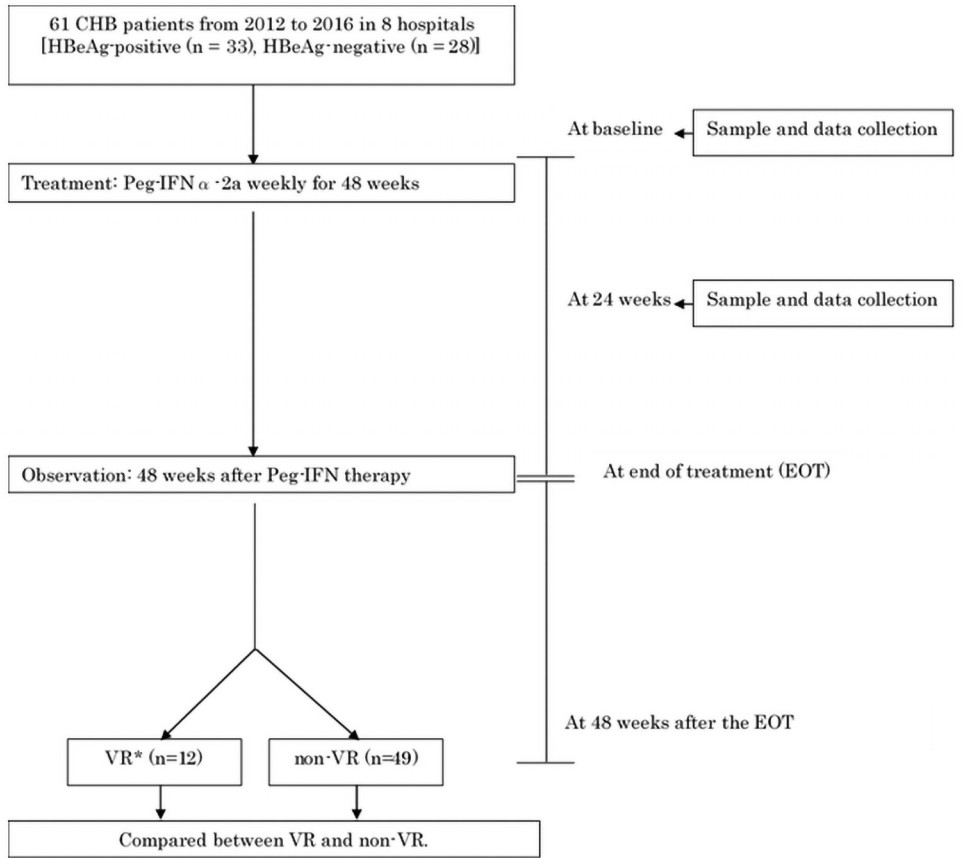

**Fig 1. Study design.** * Virological response (VR) was defined as HBV DNA < 2,000 IU/mL. HBeAg-negative, nucleos
(t)ide analogue free at the end of observation (48 weeks after the end of Peg-IFN therapy). Abbreviations: CHB,
chronic hepatitis B; HBeAg, hepatitis B e antigen; Peg-IFN, pegylated interferon; EOT, end of treatment.

primary biliary cirrhosis, or co-infection with hepatitis C virus or human immunodeficiency
virus were excluded from this study. HBeAg was positive in 33 patients and negative in 28.

The Guidelines for Hepatitis B Treatment by the Japanese Society of Hepatology recom-
mend to treat CHB patients with ALT ≥ 31 U/L and HBV DNA ≥ 2,000 IU/mL [16]. In addi-
tion, referring to several predictive factors of response to IFN therapy in CHB, such as HBeAg
status, HBV DNA, and ALT levels, the attending physicians introduced Peg-IFN therapy to
the subjects in this study. None of the patients received NAs within 48 weeks prior to Peg-IFN
treatment. Patients were treated with Peg-IFNα-2a weekly for 48 weeks and were observed for
48 weeks after the end of treatment (EOT) with monitoring at monthly intervals. At each fol-
low-up, data for biochemical markers, virological markers, blood counts, and clinical status
were recorded. Following data collection, the patients were divided into virological response
(VR) and non-VR groups. The VR was defined as ALT < 31U/L, HBV DNA < 2,000 IU/mL,
HBeAg-negative, and no need for administration of NAs until the end of the observation
period (48 weeks after the EOT). Serum samples were collected at baseline and 24-weeks dur-
ing the treatment.

Written informed consent was obtained from all individual participants. The study protocol
conformed to the ethics guidelines of the Declaration of Helsinki and was approved by the
institutional ethics review committee of Nagoya City University Hospital (60-08-0024).

We collected serum from 61 patients who received Peg-IFN monotherapy for untreated CHB. Patients were treated with Peg-IFNα-2a weekly for 48 weeks and observed up to 48 weeks after therapy. Serum samples were collected at baseline and 24-weeks during the treatment. The VR was defined as ALT < 31U/L, HBV DNA < 2,000 IU/mL, HBeAg-negative, and NAs free at 48 weeks after the EOT. There were 12 VR cases and 49 non-VR cases in this study, and comparisons were made between the VR group and the non-VR group.

## Laboratory tests and serological and virological assays

Hematologic and blood chemistry tests were carried out using standard assays. Serum HBV DNA levels were measured using COBAS TaqMan HBV 2.0 (Roche Diagnostics K. K., Tokyo, Japan [lower limit of detection, 20 IU/mL]) [17]. Positive results (signals) below the quantitative HBV DNA concentrations were referred to as "detected", which was defined as < 1.3 log IU/mL, and negative signals, as "not detected". HBeAg was determined using an HISCL HBeAg kit (Sysmex, Kobe, Japan) and HBsAg was determined using an HISCL HBsAg (Sysmex, Kobe, Japan) (detection range, 30 to 2,500,000 mIU/mL). Hepatitis B core-related antigen (HBcrAg) was determined by Lumipulse HBcrAg assay (Fujirebio.K.K., Tokyo, Japan) (detection range, 3.0 to 6.7 log U/mL). The genotypes of HBV were determined serologically by enzyme immunoassay using commercial kits, HBV GENOTYPE EIA (Institutes of Immunology Co., LTD, Tokyo, Japan).

**Sampling serum and isolation of RNA.** We followed the protocols of sampling serum and isolation of RNA as we described previously [18]. Peripheral blood was collected from each participant at baseline and during 24-week treatment and was centrifuged at 1,500 g for 5 minutes at room temperature. After serum separation, the samples were stored at -80˚C until use. Total RNAs including miRNAs in serum were purified with miRNeasy Serum/Plasma kits (Qiagen, Hilden, Germany) following the manufacturer's instructions. Specifically, we extracted total RNA from 200 μL of serum from each subject, to which $5.6 \times 10^8$ copies of *Caenorhabditis elegans* cel-miR-39-3p (cel-miR-39-3p) were added as spike-in RNA for later normalization; then total RNA was eluted from each column with 30 μL of nuclease-free water. The concentration of total RNA was quantified using a NanoDrop 2000c spectrophotometer (Thermo Fisher Scientific, Waltham, Massachusetts, USA).

**Measurement of serum miRNAs.** The levels of miRNA levels were determined using by quantitative real-time polymerase chain reaction (qRT-PCR) with Step One Plus (Thermo Fisher Scientific, Waltham, Massachusetts, USA) and TaqMan MicroRNA Assay: hsa-miR-192-5p (assay ID 000491), has-miR-122-3p (assay ID 002130), has-miR-320a (assay ID 002277), hsa-miR-6126-5p (assay ID 475618), and cel-miR-39-3p (assay ID 000200) (Thermo Fisher Scientific). One microliter of total RNA extracted from serum were subjected to reverse transcription with a TaqMan MicroRNA Reverse Transcription Kit (Thermo Fisher Scientific) and the respective TaqMan MicroRNA Assay reagents for the target molecules, in a total volume of 15 μL, followed by qRT-PCR in a total volume of 20 μL, according to the manufacturer's protocol. Amplification was carried out as follows: 95˚C for 10 min, 45 cycles at 95˚C for 15 s and 60˚C for 60 s. All reactions were carried out in duplicate. Cycle threshold (Ct) values were calculated using Step One Software v2.3 (Thermo Fisher Scientific). Expression levels of miRNAs were normalized to those of the spike-in cel-miR-39-3p. The expression levels were determined by the $2^{-\Delta Ct}$ method, in which ΔCt was calculated as: ΔCt = Ct (miRNA (miR-192-5p, miR-122-3p, miR-320a and miR-6126-5p)–Ct (cel-miR-39-3p)).

## Statistical analysis

Categorical variables were compared between groups by chi-square test, and non-categorical variables were analyzed by Mann–Whitney *U* test. Changes in serum miRNA levels from

baseline to 24-weeks were compared by two-way analysis of variance. Receiver operating characteristic (ROC) curve analyses were carried out and the AUC was calculated to evaluate the feasibility of using the miRNA levels as markers for discriminating VR. Multivariate logistic regression analyses were performed to determine whether several covariates were independently associated with VR. A *P value* < 0.05 was considered significant in all tests was set as the target variables in the multiple regression formula for in multivariate analysis. Correlation coefficients were calculated using Pearson's correlation test. Statistical analyses were performed using BellCurve Excel statistics (SSRI Inc., Tokyo, Japan).

## Results

### Comparison of clinical characteristics at baseline according to the efficacy of Peg-IFN therapy

The clinical characteristics between the VR and the non-VR groups at baseline are shown in Table 1. At the end of the observation period, 12 patients had a VR and 49 had non-VR. Compared with the VR group, the non-VR group had a significantly higher HBeAg-positive rate and levels of ALT, HBV DNA, and serum miR-192-5p at baseline ($P = 0.024$, $P = 0.020$, $P = 0.007$, $P = 0.021$, respectively). It has been reported that HBeAg-negative patients at baseline were successfully treated with Peg-IFN compared to HBeAg-positive patients at baseline [19, 20]. Therefore, we also compared clinical characteristics between the VR and non-VR groups among HBeAg-negative patients, which showed that there was no significant difference in ALT and HBV DNA levels, while miR-192-5p levels tended to be higher in the non-VR group (S1 Table). Since most patients had HBV genotype C in this study, there was no significant difference of HBV genotypes between the VR and non-VR groups.

**Table 1. Clinical characteristics of chronic Hepatitis B patients and comparison between VR group and non-VR group at baseline.**

| Factor | Total (n = 61) | VR group (n = 12) | non-VR group (n = 49) | P-value |
|---|---|---|---|---|
| Age, years | 35 (31–42) | 42 (31–46) | 35 (31–39) | 0.154 |
| Male, n (%) | 35 (57) | 5 (42) | 30 (61) | 0.367 |
| HBV genotype A/B/C | 4 / 6 / 51 | 2 / 1 / 9 | 2 / 5 / 42 | 0.287 |
| AST (U/L) | 47 (28–102) | 38 (22–69) | 49 (31–103) | 0.178 |
| ALT (U/L) | 79 (38–171) | 32 (26–94) | 85 (40–182) | 0.020 |
| Platelet counts (×10$^9$/L) | 198 (166–224) | 215 (182–220) | 197 (166–226) | 0.508 |
| FIB-4 index (C.O.I) | 1.13 (0.76–1.41) | 1.24 (1.09–1.33) | 1.00 (0.72–1.48) | 0.330 |
| HBeAg-positive, n (%) | 33 (54) | 3 (25) | 30 (61) | 0.024 |
| HBV DNA (log IU/mL) | 6.2 (4.7–8.0) | 4.3 (3.5–6.2) | 7.1 (5.0–8.2) | 0.007 |
| HBsAg (IU/mL) | 7,989 (2,290–15,940) | 3,444 (1,665–9,912) | 10,470 (2,868–23,493) | 0.066 |
| HBcrAg (log U/mL) | 5.7 (4.0–6.9) | 4.2 (2.9–6.8) | 6.0 (4.4–6.9) | 0.114 |
| miR-192-5p | 0.032 (0.016–0.086) | 0.016 (0.007–0.032) | 0.048 (0.020–0.123) | 0.021 |
| miR-320a | 0.229 (0.148–0.296) | 0.185 (0.128–0.209) | 0.249 (0.156–0.309) | 0.066 |
| miR-122-3p | 0.002 (< 0.001–0.007) | 0.003 (< 0.001–0.007) | 0.002 (0.001–0.006) | 0.899 |
| miR-6126-5p | 0.112 (0.053–0.205) | 0.057 (0.036–0.182) | 0.119 (0.067–0.205) | 0.101 |

Data from all patients were expressed as numbers for categorical data and medians (first–third quartiles) for noncategorical data.

Categorical variables were compared between groups by the chi-square test, and noncategorical variables were compared using the Mann-Whitney *U* test.

Abbreviations: VR, virological response; HBV, hepatitis B virus; AST, aspartate transaminase; ALT, alanine transaminase; FIB-4, fibrosis-4; HBsAg, hepatitis B surface antigen; HBcrAg, hepatitis B core-related antigen.

### Predictive factors for the efficacy of Peg-IFN therapy during the treatment

The clinical characteristics of the VR and non-VR groups at 24-weeks during the treatment are shown in Table 2. The proportion of HBeAg-positive patients and the levels of HBV DNA and serum miR-192-5p were significantly higher in the non-VR versus the VR group ($P = 0.020$, $P < 0.001$, $P = 0.011$, respectively).

### Transition of serum miR-192-5p levels from baseline to 24-weeks during the treatment

The transition of miR-192-5p levels from baseline to 24-weeks during the treatment is shown in S1 Fig. The levels of serum miR-192-5p were lower at baseline and 24-weeks in the VR group than the non-VR group.

### Cutoff values of variables for predicting virological response to Peg-IFN therapy

The ROC curves were created using factors that showed significant differences in the comparison between the two groups at baseline and 24-weeks, and the cutoff value and area under the curve (AUC) were calculated (Fig 2). The cutoff values for miR-192-5p at baseline and 24-weeks were 0.0159 and 0.0158, respectively. And that for HBV DNA at baseline was 4.74 log IU/ml. The AUCs for miR-192-5p levels at baseline and 24-weeks, and HBV DNA levels at baseline were 0.72, 0.74, and 0.75, respectively, indicating that they were equivalent in predicting the response to Peg-IFN therapy (Table 3).

### Prediction of VR to Peg-IFN therapy using miR-192-5p and HBV DNA levels

We calculated the sensitivity, specificity, and positive and negative predictive values (PPV and NPV) for predicting VR, using the above-mentioned cutoff values for miR-192-5p levels at baseline and 24-weeks (Table 4). The PPVs were 44% and 38%, and the NPVs were 89% and 90%, using the cutoff values for miR-192-5p levels at baseline and 24-weeks respectively, indicating that they were equivalent in predicting the response to IFN therapy. The PPV and NPV

**Table 2. Comparison of clinical data between VR and non-VR groups at 24-weeks during Peg-IFN therapy.**

| Factor | Total (n = 61) | VR group (n = 12) | non-VR group (n = 49) | P-value |
|---|---|---|---|---|
| AST (IU/L) | 34 (27–50) | 32 (22–43) | 36 (27–51) | 0.419 |
| ALT (IU/L) | 43 (27–64) | 32 (20–45) | 45 (29–67) | 0.152 |
| HBV DNA (log IU/mL) | 2.9 (< 1.3–5.0) | < 1.3 (ND–1.3) | 3.7 (1.6–5.9) | < 0.001 |
| HBsAg (IU/mL) | 2,800 (957–8,490) | 1,668 (60–3,035) | 3,392 (1,019–9,977) | 0.072 |
| HBeAg-positive, n (%) | 30 (49) | 2 (17) | 28 (57) | 0.020 |
| HBcrAg (log U/mL) | 5.4 (3.3–6.6) | 3.8 (3.2–5.1) | 6.0 (3.5–6.7) | 0.064 |
| miR-192-5p | 0.024 (0.011–0.045) | 0.012 (0.005–0.025) | 0.027 (0.013–0.061) | 0.011 |
| miR-320a | 0.198 (0.118–0.267) | 0.123 (0.090–0.216) | 0.210 (0.140–0.269) | 0.093 |
| miR-122-3p | 0.001 (< 0.001–0.004) | 0.001 (< 0.001–0.001) | 0.001 (< 0.001–0.005) | 0.121 |
| miR-6126-5p | 0.056 (0.034–0.102) | 0.045 (0.018–0.092) | 0.061 (0.034–0.104) | 0.420 |

Data from all patients were expressed as numbers for categorical data and medians (first–third quartiles) for noncategorical data.

Categorical variables were compared between groups by the chi-square test, and noncategorical variables were compared using the Mann-Whitney *U* test. Positive result (signal) below the quantitative HBV DNA concentrations was described as "<1.3", and negative signal was described as "ND".

Abbreviations: Peg-IFN, pegylated interferon; ND, not detected.

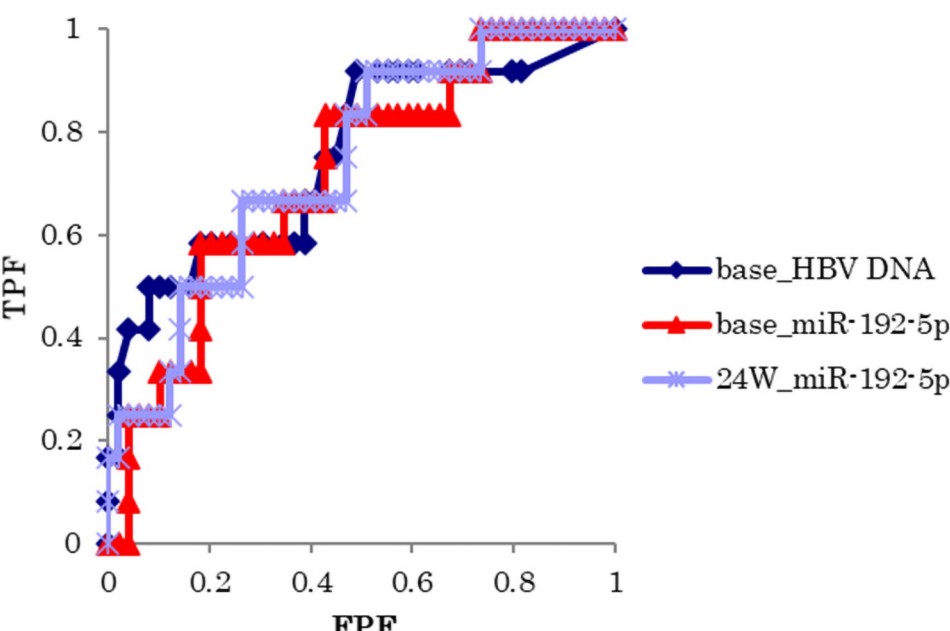

**Fig 2. ROC curves for HBV DNA and miR-192-5p.** The ROC curves were created using, miR-192-5p at baseline and 24-weeks, and HBV DNA at baseline as factors. Cutoff values were calculated from the point on the ROC curve closest to the top left of HBV DNA at baseline, miR-192-5p at baseline and 24-weeks. Abbreviations: TPF, true positive fraction; FPF, false positive fraction.

using the cutoff value for HBV DNA levels at baseline were 43% and 87%, respectively. Furthermore, combination of miR-192-5p < 0.0159 and HBV DNA < 4.74 at baseline could improve the PPV (71%), specificity (96%), and accuracy (85%). When we stratified the patients according to the cut-off values of serum HBV DNA and miR-192-5p levels at baseline, the VR rates were 57% in patients with HBV DNA < 4.74 and miR-192-5p < 0.0159; 29% in those with HBV DNA < 4.74 and miR-192-5p ≥ 0.0159; 22% in those with HBV DNA ≥ 4.74 and miR-192-5p < 0.0159; 11% in those with HBV DNA ≥ 4.74 and miR-192-5p ≥ 0.0159, respectively (S2 Fig). These data indicate that combining data of serum HBV DNA and miR-192-5p levels could improve the ability to predict the response to Peg-IFN therapy for CHB.

**Table 3. Area under the curve and cut-off values for predicting virological response to Peg-IFN therapy.**

| | | 95%CI | | | |
|---|---|---|---|---|---|
| Factor | Area under the curve | lower | higher | *P*-value | cutoff value |
| HBV DNA at baseline | 0.75 | 0.58 | 0.92 | 0.004 | 4.74 |
| miR-192 at baseline | 0.72 | 0.56 | 0.88 | 0.007 | 0.0159 |
| miR-192 at 24-weeks | 0.74 | 0.59 | 0.89 | 0.002 | 0.0158 |

Area under the curve was calculated from the receiver operating characteristic (ROC) curve as shown in Fig 2.

The cutoff values were calculated from the point on the ROC curve closest to top left of HBV DNA at baseline and miR-192-5p levels at baseline and 24-weeks during Peg-IFN therapy.

Abbreviations: CI, confidence interval.

**Table 4. Prediction of virological response to Peg-IFN therapy.**

|  | PPV | NPV | sensitivity | specificity | accuracy |
|---|---|---|---|---|---|
| miR-192-5p < 0.0159 at baseline | 7 / 16 (44%) | 40 / 45 (89%) | 7 / 12 (58%) | 40 / 49 (82%) | 47 / 61 (77%) |
| miR-192-5p < 0.0158 at 24-weeks | 8 / 21 (38%) | 36 / 40 (90%) | 8 / 12 (67%) | 36 / 49 (73%) | 44 / 61 (72%) |
| HBV DNA < 4.74 at baseline | 6 / 14 (43%) | 41 / 47 (87%) | 6 / 12 (50%) | 41 / 49 (84%) | 47 / 61 (77%) |
| HBV DNA < 4.74 and miR-192-5p < 0.0159 at baseline | 5 / 7 (71%) | 47 / 54 (87%) | 5 / 12 (42%) | 47 / 49 (96%) | 51 / 61 (85%) |

The cut-off values of HBV DNA level at baseline and the miR-192 levels at baseline and 24-weeks during the Peg-IFN therapy for predicting virological response were determined by the ROC analyses.

Abbreviations: PPV, positive predictive value; NPV, negative predictive value.

## Multivariate logistic regression analysis for predicting virological response to Peg-IFN therapy

Next, we conducted multivariate logistic regression analysis to determine predictive factors for discriminating VR. Based on previous findings regarding predictive factors for the response to IFN therapy in CHB patients as described in the introduction [9, 10], and our results as shown in Tables 1–4, we included the following variables as covariates: HBeAg, HBV DNA level < 4.74 log IU/ml and miR-192-5p level < 0.0159 at baseline. In this study, unexpectedly, ALT levels at baseline were higher in the non-VR group than the VR group due to the small number of cases (Table 1), although several high quality studies showed that higher ALT level was a predictor of the good response to Peg-IFN therapy in CHB patients with or without HBeAg [8–10]. Therefore, considering sampling errors due to the small number of cases, we did not include ALT as a covariate in the analysis. As a result, miR-192-5p level < 0.0159 at baseline was identified an independent predictive factor of VR (odds ratio = 4.5; $P$ = 0.041) (Table 5).

## Relationship of miR-192-5p expression levels in serum with clinical parameters

We examined the correlations between serum miR-192-5p levels and other clinical parameters, which showed significant correlations with the levels of HBV DNA, HBcrAg, and especially HBsAg, at baseline and 24-weeks, whereas there was no correlation with the levels of AST, ALT, and platelet counts (Table 6). In addition, serum miR-192-5p levels at baseline were significantly higher in HBeAg-positive versus than HBeAg-negative patients (0.063 vs. 0.022, $P$ = 0.0058), but there was no difference at 24-weeks between the two groups.

**Table 5. Multivariate logistic regression analysis of factors at baseline associated with virological response to Peg-IFN therapy.**

| | | | 95% CI | |
|---|---|---|---|---|
| Factor | *P*-value | Odds | Lower | Upper |
| HBeAg-negative | 0.198 | 2.9 | 0.57 | 14.61 |
| HBV DNA < 4.74* | 0.310 | 2.3 | 0.46 | 11.12 |
| miR-192-5p < 0.0159* | 0.041 | 4.5 | 1.06 | 19.20 |

* The cut-off values of HBV DNA and miR-192 levels at baseline for predicting virological response was determined by the ROC analysis.

**Table 6. Correlations of miR-192-5p expression levels in serum with other clinical parameters.**

| | miR-192-5p levels | | | |
| | r* | | *P*-value | |
| Factor | baseline | 24-weeks | baseline | 24-weeks |
|---|---|---|---|---|
| AST | -0.055 | -0.080 | 0.700 | 0.568 |
| ALT | 0.082 | 0.092 | 0.566 | 0.513 |
| PLT | -0.096 | N.A. | 0.467 | N.A. |
| HBV DNA | 0.484 | 0.655 | < 0.001 | < 0.001 |
| HBsAg | 0.759 | 0.730 | < 0.001 | < 0.001 |
| HBcrAg | 0.384 | 0.551 | 0.005 | < 0.001 |

* The correlation coefficient (r) is calculated using the Pearson correlation test.

Abbreviations: PLT, platelet counts; N.A., not available.

## Discussion

The present study validated that serum miR-192-5p levels were associated with response to IFN therapy in CHB patients, whereas serum miR-320a, miR-122-3p, and miR-6129-5p levels were not validated. Multivariate analysis showed that a lower miR-192-5p level at baseline was an independent predictor of VR. To date, several factors have been reported to be associated with VR in the treatment of CHB, for instance patients with HBeAg-negative CHB who show decreased HBcrAg levels during treatment with Peg-IFN combination therapy with or without NAs were more likely to succeed with anti-viral treatment [21, 22]. Another study in HBeAg-negative CHB patients mostly with genotype D, showed that no reduction in HBsAg levels and a lack of decrease in HBV DNA levels below 1.2 $\log_{10}$ IU/ml during 12-week treatment has a NPV of nearly 100% of untreated persistent VR, and this was therefore recommended as a stopping rule for early discontinuation of ineffective Peg-IFN [20]. In addition, several studies have indicated that serum HBV DNA level is a useful and reliable marker to predict the efficacy of Peg-IFN therapy for CHB [9, 10]. The present study showed that the AUC value of serum HBV DNA levels at baseline for predicting VR was the best. Therefore, we think that serum miR-192-5p level cannot replace HBV DNA, but improve to predict the response to Peg-IFN therapy for CHB. Actually, combining data of serum miR-192-5p and HBV DNA levels at baseline could improve the ability to predict the response to Peg-IFN therapy (Table 4 and S2 Fig).

MicroRNA-192-5p is associated with hypertension, diabetes, and various cancers, such as lung, gastric, pancreatic, and liver cancer, and its potential as a disease marker has been suggested [23–28]. It has been reported that miR-192-5p is abundantly expressed in hepatic tissues [29], as well as in serum and urine, and their exosome [30]. Several studies on liver diseases have identified serum miR-192-5p level as a potential early biomarker for detecting hepatocellular carcinoma (HCC) [31], and it was revealed that miR-192-5p plays an important role in the pathophysiology of non-alcoholic fatty liver disease (NAFLD) and liver injury [32]. As for HBV-related diseases, the replication of HBV was shown to be correlated with the in vitro expression of miR-192-5p in a HepG2 cell model system, and overexpression of miR-192-5p by mimics reduces the protein level of pro-apoptotic BIM (Bcl-2-like protein 11) [33]. In addition, miR-192-5p was shown to be overexpressed in both the sera and HBsAg particles of CHB patients [14]. Subsequent research revealed that miR-192-5p is present in hepatoma-derived extracellular vesicles and abundantly expressed in HBeAg-positive patients compared with HBeAg-negative patients [34]. Our study showed a strong correlation between HBsAg and miR-192-5p levels in serum (Table 6), and miR-192-5p levels were higher in HBeAg-positive

patients than in HBeAg-negative patients, while the VR rate tended to be lower in HBeAg-positive versus HBeAg-negative patients (3/33 vs. 9/28, $P = 0.053$) (S2 Table). Thus, HBV replication might influence serum miR-192-5p levels, which accounts for serum miR-192-5p levels being associated with the response to IFN therapy. However, serum miR-192-5p level at baseline was an independent predictor for VR in our study; therefore, there might be a further mechanism by which miR-192-5p influences or predicts IFN efficacy. Intriguingly, previous studies demonstrated induction of miR-192 by IFN-α in Huh7.5 cells and downregulation upon hepatitis C virus infection [35], indicating that miR-192-5p upregulation in serum could be an independent variable for non-response to Peg-IFN and ribavirin treatment in chronic hepatitis C [36]. Additionally, miR-192-5p was increased in serum exosome from NAFLD patients, and hepatocyte-derived exosomal miR-192-5p promoted the polarization of inflammatory macrophage which induced immune response and inflammation [37]. These findings led us to speculate that miR-192-5p might be associated with antiviral immunity. Meanwhile, the mechanism that miR-192-5p is secreted in peripheral blood from HCC cells has not been elucidated. Further studies are necessary to elucidate the mechanism of secretion and functional roles of miR-192-5p in CHB and other liver diseases.

There were several limitations to this study. First, ALT levels were low in the VR cases, although previous studies have reported that ALT levels were high in responders to IFN therapy for CHB [8–10, 38]. In the non-VR group, higher HBV DNA levels and HBeAg-positive rate, which were considered to be predictors for non-response to IFN therapy in CHB, were associated with active hepatic inflammation, namely higher ALT levels. Second, we examined only a small number of patients, especially in the VR group. This resulted from difficulty in collecting VR cases, because few CHB patients being treated with Peg-IFN and the low VR rate. Third, the majority of the patients had HBV genotype B or C. Fourth, we carried out elastography or liver biopsy in less than half of the subjects, therefore, we have the date on hepatic fibrosis grade using FIB-4 index at baseline only. FIB-4 index is calculated using age, AST, ALT and PLT levels. Since IFN therapy affects AST, ALT and platelet levels, FIB-4 index might not be appropriate to evaluate the changes of hepatic fibrosis grade during and after IFN therapy. Therefore, we could not evaluate the efficacy in hepatic fibrosis grade after the initiation of Peg-IFN therapy. Future study is needed to evaluate serum miR-192-5p level as a marker for the response to Peg-IFN therapy in a large group of CHB patients with various HBV genotypes, who should be divided by HBeAg status.

In conclusion, serum miR-192-5p levels might be a predictive biomarker for the response to Peg-IFN therapy in CHB patients. Our results and previous findings let us speculate that miR-192-5p might be associate with HBV replication and antiviral immunity. Further studies are necessary to elucidate the functional roles of miR-192-5p in CHB.

## Supporting information

**S1 Fig. Transition of serum miR-192-5p levels from baseline to 24-weeks during the treatment in VR, non-VR and all patients.**
(TIF)

**S2 Fig. Virological response rates stratified by the cut-off values of serum HBV DNA and miR-192-5p levels at baseline.**
(TIF)

**S1 Table. Comparison of clinical characteristics of HBeAg-negative patients between VR and non-VR groups.**
(DOCX)

**S2 Table. Comparison of clinical characteristics of patients between HBeAg-positive and HBeAg-negative.**
(DOCX)

## Acknowledgments

We appreciate Noboru Shinkai, Shuko Murakami and Kyoko Ito for supporting the analyses.

## Author Contributions

**Conceptualization:** Yoshihito Nagura, Kentaro Matsuura, Yasuhito Tanaka.

**Data curation:** Yoshihito Nagura.

**Formal analysis:** Yoshihito Nagura.

**Funding acquisition:** Yasuhito Tanaka.

**Resources:** Kentaro Matsuura, Etsuko Iio, Koji Fujita, Takako Inoue, Akihiro Matsumoto, Eiji Tanaka, Shuhei Nishiguchi, Jong-Hon Kang, Takeshi Matsui, Masaru Enomoto, Hiroki Ikeda, Tsunamasa Watanabe, Chiaki Okuse, Masataka Tsuge, Masanori Atsukawa, Masakuni Tateyama, Hiromi Kataoka, Yasuhito Tanaka.

**Supervision:** Yasuhito Tanaka.

**Validation:** Yoshihito Nagura.

**Writing – original draft:** Yoshihito Nagura.

**Writing – review & editing:** Kentaro Matsuura, Yasuhito Tanaka.

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
