## [Decision Letter · Decision Letter 0]

24 Nov 2021

PONE-D-21-35260Serum miR-192-5p Levels Predict the Efficacy of Pegylated Interferon Therapy for Chronic Hepatitis BPLOS ONE

Dear Dr. Tanaka,

Thank you for submitting your manuscript to PLOS ONE. After careful consideration, we feel that it has merit but does not fully meet PLOS ONE’s publication criteria as it currently stands. Therefore, we invite you to submit a revised version of the manuscript that addresses the points raised by two reviewers during the review process. Please submit your revised manuscript by Jan 08 2022 11:59PM. If you will need more time than this to complete your revisions, please reply to this message or contact the journal office at plosone@plos.org. Please include the following items when submitting your revised manuscript:A rebuttal letter that responds to each point raised by the academic editor and reviewer(s). You should upload this letter as a separate file labeled 'Response to Reviewers'.A marked-up copy of your manuscript that highlights changes made to the original version. You should upload this as a separate file labeled 'Revised Manuscript with Track Changes'.An unmarked version of your revised paper without tracked changes. You should upload this as a separate file labeled 'Manuscript'.

We look forward to receiving your revised manuscript.

Kind regards,

Wenyu Lin, PhD

Academic Editor

PLOS ONE

Journal Requirements:

Reviewers' comments:

Reviewer's Responses to Questions

**Comments to the Author**

1. Is the manuscript technically sound, and do the data support the conclusions?

Reviewer #1: Partly

Reviewer #2: Yes

2. Has the statistical analysis been performed appropriately and rigorously? 

Reviewer #1: Yes

Reviewer #2: No

3. Have the authors made all data underlying the findings in their manuscript fully available?

Reviewer #1: Yes

Reviewer #2: Yes

4. Is the manuscript presented in an intelligible fashion and written in standard English?

Reviewer #1: Yes

Reviewer #2: Yes

5. Review Comments to the Author

Reviewer #1: 1. HBV-DNA is a reliable and convenient biomarker for virological response. Therefore, miR-192-5p level, a not routine biomarker, is useless to virological response.

2. The levels of serum ALT and HBV DNA as well as liver fibrosis are important predictors of long-term outcome that inform decisions for treatment initiation as well as treatment response. Additionally, staging of liver disease severity using liver biopsy or noninvasive tests such as elastography are important in guiding surveillance and assisting with treatment decisions. Therefore, authors should better consider those factors changes during the antiviral treatment to indicate the efficacy of pegylated interferon therapy, not only virological response.

Reviewer #2: Nagura Y, et al. reported the association between serum miRNA levels (especially miRNA-192-5p) and the efficacy of Peg-IFN monotherapy for chronic hepatitis B patients. These data are very interesting and might be new important knowledge, however there were some concerns as reviewer describes below.

Major points:

1. In Table1, AST or ALT levels of your patients are relatively low, how did you think about treatment indication of Peg-IFN therapy?

2. The outcome in this study was determined by virological response which was defined as HBV DNA < 2000 IU/mL, HBeAg-negative, and no need for administration of NAs.

This reviewer think normalization of transaminase should be included in treatment response.

3. Authors described the cutoff values for miR-192-5p levels at baseline and 24-week were same (0.016). Is this result means that miR-192-5p levels did not changed before and after Peg-IFN therapy in each patient? Was the response of miR-192-5p levels between before and 24-week after Peg-IFN therapy factor of treatment response?

4.The ALT level is significantly different between VR and non-VR group in Table1. You should include the input covariates in multivariate analysis in Table 5.

5. The HBsAg level seems to be different between VR and non-VR group although there is not significant different. Is the Mann-Whitney U test eligible method to this factor?

6. In discussion section reference 29 is not reported about serum miR-192-5p. You should change this paper to original report about miR-192-5p for detecting in hepatocellular carcinoma. In past reports miR-192-5p is "upregulated" in hepatocellular carcinoma or non-alcoholic fatty liver disease, however about response of Peg-IFN therapy miR-192-5p is "lower" in virological-response group in this study. Please discuss about this phenomenon.

Minor points:

1. You should describe the definition of outcome in this study in the method section not in the result section.

2. Please show clarify miR-192-5p is measured at what time point (baseline, 24 weeks or EOT) in each description.

3. In the section “Correlations of miR-192-50 expression levels in serum with clinical parameters”, the result of HBeAg-positive or negative is not eligible in this title.

6. PLOS authors have the option to publish the peer review history of their article (what does this mean?). If published, this will include your full peer review and any attached files.

Reviewer #1: No

Reviewer #2: No

---

## [Author Response · Author response to Decision Letter 0]

31 Dec 2021

Reviewer #1

Comment 1:

HBV-DNA is a reliable and convenient biomarker for virological response. Therefore, miR-192-5p level, a not routine biomarker, is useless to virological response.

Response:

As the reviewer pointed out, HBV DNA is a useful and convenient marker, but measuring serum miRNA is not convenient for predicting response to IFN in CHB patients. Although the abilities of predicting VR using the cutoff values for miR-192-5p and HBV DNA levels at baseline were almost equivalent, combination of miR-192-5p < 0.016 and HBV DNA < 4.74 at baseline could improve the PPV (71%) and specificity (96%) as shown in the revised Table 4. Thus, serum miR-192-5p is useful for predicting the response to IFN therapy for CHB. We added the PPV, NPV, sensitivity and specificity by the combination of HBV DNA and miR-192-5p in Table 4 and the description in the text in page 14, line 303-304 and in page 18, line 373-375.

Table 4. Prediction of virological response to Peg-IFN therapy.

　 PPV NPV sensitivity specificity

miR-192-5p < 0.0159 at baseline 7 / 16 (44%) 40 / 45 (89%) 7 / 12 (58%) 40 / 49 (82%)

miR-192-5p < 0.0158 at 24-weeks 8 / 21 (38%) 36 / 40 (90%) 8 / 12 (67%) 36 / 49 (73%)

HBV DNA < 4.74 at baseline 6 / 14 (43%) 41 / 47 (87%) 6 / 12 (50%) 41 / 49 (84%)

HBV DNA < 4.74 and miR-192-5p < 0.0159 at baseline 5 / 7(71%) 47 / 54 (87%) 5 / 12 (42%) 47 / 49 (96%)

Comment 2:

The levels of serum ALT and HBV DNA as well as liver fibrosis are important predictors of long-term outcome that inform decisions for treatment initiation as well as treatment response. Additionally, staging of liver disease severity using liver biopsy or noninvasive tests such as elastography are important in guiding surveillance and assisting with treatment decisions. Therefore, authors should better consider those factors changes during the antiviral treatment to indicate the efficacy of pegylated interferon therapy, not only virological response.

Response:

Thank you for your important comment. Unfortunately, we carried out elastography or liver biopsy in less than half of the subjects, therefore, we have the date on hepatic fibrosis grade using FIB-4 index at baseline only (Table 1), which showed that those levels at baseline were not significantly different between VR and non-VR. As well known, the FIB-4 index is calculated using age, AST, ALT and PLT levels. Since IFN therapy affects AST, ALT and PLT levels, FIB-4 index might not be appropriate to evaluate the changes of hepatic fibrosis during and after IFN therapy in CHB. Therefore, we could not evaluate the efficacy in hepatic fibrosis grade after the initiation of IFN therapy. We added the description as a limitation in the discussion section, in page 20, line 418-424.

Reviewer #2

Major comment 1:

In Table1, AST or ALT levels of your patients are relatively low, how did you think about treatment indication of Peg-IFN therapy?

Response:

The Guidelines for Hepatitis B Treatment by the Japanese Society of Hepatology recommends to treat CHB patients with ALT ≥ 31 U/L and HBV DNA ≥ 2,000 IU/mL (1). In addition, referring to several predictive factors of response to IFN therapy in CHB, such as HBeAg status, HBV DNA, and ALT levels, the attending physicians introduced Peg-IFN therapy in the subjects in this study. In some patients, ALT levels were not met the above criteria at baseline, but their ALT levels were met it by multiple blood tests before IFN therapy. We added this description in page 5-6, line 119-123.

Major comment 2：

The outcome in this study was determined by virological response which was defined as HBV DNA < 2000 IU/mL, HBeAg-negative, and no need for administration of NAs.

This reviewer think normalization of transaminase should be included in treatment response.

Response:

Thank you for your important suggestion. We defined a virological response as “ALT < 31U/L, HBV DNA < 2,000 IU/mL, HBeAg-negative, and no need for administration of NAs until the end of the observation period”. We revised the description in page 6, line 129-131.

Major comment 3：

Authors described the cutoff values for miR-192-5p levels at baseline and 24-week were same (0.016). Is this result means that miR-192-5p levels did not changed before and after Peg-IFN therapy in each patient? Was the response of miR-192-5p levels between before and 24-week after Peg-IFN therapy factor of treatment response?

Response:

The cutoff values of miR-192-5p at baseline and 24 weeks were 0.0159 and 0.0158, respectively. We revised these data of the cut-off values in the text and Table 3-5. 

In addition, we showed the changes of miR-192-5p levels from baseline to 24-weeks in VR and non-VR groups as below. The miR-192-5p levels were lower at baseline and 24 weeks in the VR group than the non-VR group. We add these data in the text in page 13, line 270-272 and the supplementary file. 

Major comment 4：

The ALT level is significantly different between VR and non-VR group in Table1. You should include the input covariates in multivariate analysis in Table 5.

Response:

Based on previous findings regarding predictive factors of the response to IFN therapy in CHB patients as described in the introduction (2) (3) (4), and our results as shown in Tables 1-4, we selected covariates in the multivariate logistic regression analysis. In this study, ALT levels at baseline were higher in the non-VR group than the VR group (Table 1). But, several high quality studies showed that higher ALT level was a predictor of the good response to Peg-IFN treatment in CHB patients with or without HBeAg (2) (3) (4). Therefore, we did not include ALT as a covariate in the analysis. We added the descriptions in page 15-16, line 323-325, and described regarding higher ALT levels in the non-VR group as a limitation in the discussion in page 19-20, line 410-415.

Major comment 5：

The HBsAg level seems to be different between VR and non-VR group although there is not significant different. Is the Mann-Whitney U test eligible method to this factor?

Response:

Since almost all parameters including HBsAg in Table 1 and 2 were not normally distributed, the Mann-Whitney U test, which is a nonparametric test, was appropriate to compare these parameters between the VR and non-VR groups.

Major comment 6：

In discussion section reference 29 is not reported about serum miR-192-5p. You should change this paper to original report about miR-192-5p for detecting in hepatocellular carcinoma. In past reports miR-192-5p is "upregulated" in hepatocellular carcinoma or non-alcoholic fatty liver disease, however about response of Peg-IFN therapy miR-192-5p is "lower" in virological-response group in this study. Please discuss about this phenomenon.

Response:

Thank you for your comments. We cited the original report by Zhou J, Plasma microRNA panel to diagnose hepatitis B virus-related hepatocellular carcinoma. J Clin Oncol. 2011;29(36):4781-8 (5) in page 18, line 381. As we described in the discussion, HBV replication might influence serum miR-192-5p levels, which accounts for serum miR-192-5p levels being associated with the response to IFN therapy. The mechanism that miR-192-5p is secreted in peripheral blood from HCC cells has not been elucidated. Meanwhile, miR-192-5p was increased in serum exosome in NAFLD patients, and hepatocyte-derived exosomal miR-192-5p promoted the polarization of inflammatory macrophage which induce immune response and inflammation (6). Therefore, we speculate that miR-192-5p might be associated with antiviral immunity. We revised the discussion in page 19, line 402-409.

Minor comment 1:

You should describe the definition of outcome in this study in the method section not in the result section.

Response:

Thank you for your comment. We described the definition of outcome in the Method section in page 6, line 129-132. 

Minor comment 2:

Please show clarify miR-192-5p is measured at what time point (baseline, 24 weeks or EOT) in each description.

Response:

Thank you for your comment. We used the serum samples at baseline and 24-weeks during the IFN-therapy. We revised the description regarding sample collection in the legend of Figure 1, and added it in the text in page 6, line 141-142. And we added the time point of measuring miR-192-5p in each description.

Minor comment 3:

In the section “Correlations of miR-192-5p expression levels in serum with clinical parameters”, the result of HBeAg-positive or negative is not eligible in this title.

Response:

Thank for your comment. We changed the title “Correlations of miR-192-5p expression levels in serum with HBeAg-positive or negative” To “Relationship of miR-192-5p expression levels in serum with other clinical parameters” in page 16, line 338.

Other revision

In Table 5, we corrected the factors: “HBeAg-positive” to “HBe-negative”; “HBV DNA ≥ 4.74” to “HBV DNA < 4.74”; “miR-192-5p ≥ 0.0159” to “miR-192-5p < 0.0159” to be easy to understand. As a result, the odds of these factors were changed, but their P-values were not changed.

References:

1. Drafting Committee for Hepatitis Management Guidelines tJSoH. Japan Society of Hepatology Guidelines for the Management of Hepatitis B Virus Infection: 2019 update. Hepatol Res. 2020;50(8):892-923.

2. Piratvisuth T, Lau G, Chao YC, Jin R, Chutaputti A, Zhang QB, et al. Sustained response to peginterferon alfa-2a (40 kD) with or without lamivudine in Asian patients with HBeAg-positive and HBeAg-negative chronic hepatitis B. Hepatol Int. 2008;2(1):102-10.

3. Buster EH, Hansen BE, Lau GK, Piratvisuth T, Zeuzem S, Steyerberg EW, et al. Factors that predict response of patients with hepatitis B e antigen-positive chronic hepatitis B to peginterferon-alfa. Gastroenterology. 2009;137(6):2002-9.

4. Bonino F, Marcellin P, Lau GK, Hadziyannis S, Jin R, Piratvisuth T, et al. Predicting response to peginterferon alpha-2a, lamivudine and the two combined for HBeAg-negative chronic hepatitis B. Gut. 2007;56(5):699-705.

5. Zhou J, Yu L, Gao X, Hu J, Wang J, Dai Z, et al. Plasma microRNA panel to diagnose hepatitis B virus-related hepatocellular carcinoma. J Clin Oncol. 2011;29(36):4781-8.

6. Liu XL, Pan Q, Cao HX, Xin FZ, Zhao ZH, Yang RX, et al. Lipotoxic Hepatocyte-Derived Exosomal MicroRNA 192-5p Activates Macrophages Through Rictor/Akt/Forkhead Box Transcription Factor O1 Signaling in Nonalcoholic Fatty Liver Disease. Hepatology. 2020;72(2):454-69.

---

## [Decision Letter · Decision Letter 1]

17 Jan 2022

PONE-D-21-35260R1Serum miR-192-5p Levels Predict the Efficacy of Pegylated Interferon Therapy for Chronic Hepatitis BPLOS ONE

Dear Dr. Tanaka,

Thank you for submitting your manuscript to PLOS ONE. After careful consideration, we feel that it has merit but does not fully meet PLOS ONE’s publication criteria as it currently stands. Therefore, we invite you to submit a revised version of the manuscript that addresses the points raised by reviewer #1 during the review process.

We look forward to receiving your revised manuscript.

Kind regards,

Wenyu Lin, PhD

Academic Editor

PLOS ONE

Journal Requirements:

Reviewers' comments:

Reviewer's Responses to Questions

**Comments to the Author**

1. If the authors have adequately addressed your comments raised in a previous round of review and you feel that this manuscript is now acceptable for publication, you may indicate that here to bypass the “Comments to the Author” section, enter your conflict of interest statement in the “Confidential to Editor” section, and submit your "Accept" recommendation.

Reviewer #1: (No Response)

Reviewer #2: All comments have been addressed

2. Is the manuscript technically sound, and do the data support the conclusions?

Reviewer #1: Partly

Reviewer #2: Yes

3. Has the statistical analysis been performed appropriately and rigorously? 

Reviewer #1: No

Reviewer #2: Yes

4. Have the authors made all data underlying the findings in their manuscript fully available?

Reviewer #1: No

Reviewer #2: Yes

5. Is the manuscript presented in an intelligible fashion and written in standard English?

Reviewer #1: Yes

Reviewer #2: Yes

6. Review Comments to the Author

Reviewer #1: 1. HBV-DNA is a routine biomarker to evaluate the efficacy of Peg-IFN for CHB. For this reason, even if authors clear that serum miR-192-5p is a biomarker is meaningless. The author should clear whether the serum miR-192-5p can replace HBV-DNA or help HBV-DNA to improve the prediction ability for the efficacy of Peg-IFN. Therefore, authors should add fresh evidence to illustrate serum miR-192-5p is better than HBV-DNA in predicting for the efficacy of Peg-IFN, or combine miR-192-5p and HBV-DNA could improve the prediction ability than single HBV-DNA, not for miR-192-5p.

2. The area under the curve of HBV-DNA at base is 0.75, which is more than miR-192 at base (0.72) and miR-192 at 24-weeks (0.74). Those data are not indicative that miR-192-5p is a better biomarker than HBV-DNA for predicting VR. So, what is the significance purpose of this study?

3. In addition to the area under the curve, accuracy is another important index to evaluate diagnostic ability. Authors should add the accuracy of serum miR-192-5p and/or HBV-DNA for predicting VR.

Reviewer #2: (No Response)

7. PLOS authors have the option to publish the peer review history of their article (what does this mean?). If published, this will include your full peer review and any attached files.

Reviewer #1: No

Reviewer #2: No

---

## [Author Response · Author response to Decision Letter 1]

19 Jan 2022

Reviewer #1

Comment 1:

HBV-DNA is a routine biomarker to evaluate the efficacy of Peg-IFN for CHB. For this reason, even if authors clear that serum miR-192-5p is a biomarker is meaningless. The author should clear whether the serum miR-192-5p can replace HBV-DNA or help HBV-DNA to improve the prediction ability for the efficacy of Peg-IFN. Therefore, authors should add fresh evidence to illustrate serum miR-192-5p is better than HBV-DNA in predicting for the efficacy of Peg-IFN, or combine miR-192-5p and HBV-DNA could improve the prediction ability than single HBV-DNA, not for miR-192-5p.

Response:

Thank you for the important comment. As the reviewer pointed out, HBV DNA is a useful and reliable marker to predict the efficacy of Peg-IFN therapy for CHB. Actually, the present study showed that the AUC value of HBV DNA at baseline was the best. Therefore, serum miR-192-5p level cannot replace HBV DNA, but improve to predict the response to Peg-IFN therapy for CHB. As the reviewer suggested in the comment 3, we added data of accuracy using the cut-off values of HBV DNA and miR-192-5p levels in Table 4, which indicate the combination of HBV DNA and miR-192-5p levels improved accuracy as well as PPV and specificity. In addition, when we stratified the patients according to the cut-off values of serum HBV DNA and miR-192-5p levels at baseline, the VR rates were 57% in patients with HBV DNA < 4.74 and miR-192-5p < 0.0159; 29% in those with HBV DNA < 4.74 and miR-192-5p ≥ 0.0159; 22% in those with HBV DNA ≥ 4.74 and miR-192-5p < 0.0159; 11% in those with HBV DNA ≥ 4.74 and miR-192-5p ≥ 0.0159, respectively (S2 Fig). These data indicate that combining data of serum HBV DNA and miR-192-5p levels could improve the ability to predict the response to Peg-IFN therapy for CHB. We add these description in the text, in page 14-15, line 304-308, and in page 18, line 371-378.

Comment 2:

The area under the curve of HBV-DNA at base is 0.75, which is more than miR-192 at base (0.72) and miR-192 at 24-weeks (0.74). Those data are not indicative that miR-192-5p is a better biomarker than HBV-DNA for predicting VR. So, what is the significance purpose of this study?

Response:

As we described in the above response, serum miR-192-5p level could improve the ability to predict the response to Peg-IFN therapy for CHB. Furthermore, our results and previous findings let us speculate that miR-192-5p might be associate with HBV replication and antiviral immunity as we described in the discussion, in page 19, line 405-431. We added the significance of our study in the conclusion, in page 20, line 451-454. .

Comment 3:

In addition to the area under the curve, accuracy is another important index to evaluate diagnostic ability. Authors should add the accuracy of serum miR-192-5p and/or HBV-DNA for predicting VR.

Response:

Thank you for the suggestive comment. As we described in the response to the comment 1, we added the data of the accuracy of serum miR-192-5p and/or HBV-DNA for predicting VR in Table 4.

---

## [Editor Report · Decision Letter 2]

28 Jan 2022

Serum miR-192-5p Levels Predict the Efficacy of Pegylated Interferon Therapy for Chronic Hepatitis B

PONE-D-21-35260R2

Dear Dr. Tanaka,

We’re pleased to inform you that your manuscript has been judged scientifically suitable for publication and will be formally accepted for publication once it meets all outstanding technical requirements.

Kind regards,

Wenyu Lin, PhD

Academic Editor

PLOS ONE

Additional Editor Comments (optional):

The authors have adequately addressed reviewer comments. The manuscript is suitable to publish in Plos One.
---

## [Editor Report · Acceptance letter]

3 Feb 2022

PONE-D-21-35260R2 

Serum miR-192-5p Levels Predict the Efficacy of Pegylated Interferon Therapy for Chronic Hepatitis B 

Dear Dr. Tanaka:

I'm pleased to inform you that your manuscript has been deemed suitable for publication in PLOS ONE. Congratulations! Your manuscript is now with our production department. 

Kind regards, 

on behalf of

Dr. Wenyu Lin 

Academic Editor

PLOS ONE